# Tadalafil and Steroid Hormones Interactions in Adipose, Bone and Prostate Tissues: Focus on Translational Perspectives

**DOI:** 10.3390/ijms23084191

**Published:** 2022-04-11

**Authors:** Emanuela Alessandra Greco, Cristina Antinozzi, Luigi Di Luigi, Antonio Aversa, Paolo Sgrò

**Affiliations:** 1Department Unicusano, University “Niccolò Cusano”, 00166 Rome, Italy; emanuela.greco@unicusano.it; 2Department of Movement, Human and Health Sciences, “Foro Italico” University, 00135 Rome, Italy; cristina.antinozzi@uniroma4.it (C.A.); luigi.diluigi@uniroma4.it (L.D.L.); paolo.sgro@uniroma4.it (P.S.); 3Department of Experimental and Clinical Medicine, Magna Græcia University, 88100 Catanzaro, Italy

**Keywords:** tadalafil, prostate cancer, aromatase, adipocytes, bone, androgen receptors, obesity, osteoporosis

## Abstract

Tadalafil is a selective phosphodiesterase type-5 (PDE5) inhibitor that is approved for the treatment of men with erectile dysfunction (ED) and/or benign prostate hyperplasia (BPH) -associated symptoms. Besides its classical actions on PDE5 within the genitourinary tract, where the specific enzyme expression is maximal, it may exert different systemic effects. This is mainly due to the pleiotropic distribution of PDE5 enzyme throughout the human (and animal) body, where it can exert protective effects in different clinical conditions. Recently, it has been demonstrated that tadalafil may display novel actions on androgen receptor (AR) expression and activity and cytochrome P19a1 (Cyp19a1) and estrogen receptor β (ERβ) expression in different in vitro systems, such as adipose, bone and prostate cancer cells, where it can act as a selective modulator of steroid hormone production. This may determine novel potential mechanism(s) of control in pathophysiologic pathways. In this review, we summarize basic research and translational results applicable to the use of tadalafil in the treatment of obesity, bone loss and prostate cancer.

## 1. Introduction

The phosphodiesterase (PDE) enzyme superfamily consists of 11 families (PDE1–PDE11) that modulate the intracellular concentrations of the second messengers, cyclic adenosine monophosphate (cAMP) and cyclic guanosine monophosphate (cGMP) [1]. These enzymes catalyze their degradation to the inactive 5’ nucleotide monophosphates cAMP and cGMP, regulate multiple functions and have been involved in several pathophysiological processes and diseases, including cancer [2]. Consequently, PDEs control many physiological processes, whilst their altered expression, localization and function are involved in the pathogenesis of several diseases [3,4]. The over-expression and/or aberrant activation of these enzymes can promote the onset and progression of tumors [5].

In the last decades, numerous pharmaceutical compounds that selectively inhibit the catalytic activities of PDEs have been developed for the treatment of various diseases, but only PDE5 inhibitors (PDE5i) have reached clinical application, mainly for treating male erectile dysfunction (ED) [6]. PDE5is inhibit cGMP-dependent PDE5 in the penile corporal smooth muscle, and their efficacy is based on their ability to block the cGMP break-down produced by the nitric oxide (NO) -dependent activation of guanylyl cyclase [7]. Initial studies in transgender males have demonstrated that small PDE5 amounts are present in different extra-genital tissues from humans, such as skeletal muscle and heart, lung, and adrenal gland tissue [8]. Successive preclinical and clinical observations, coming from our and other research units, demonstrated that tadalafil modulates the Cyp19a1 (ARO) activity and the expression and function of the androgen receptor (AR) in lines of bone, breast, prostate, adipose and muscle cells, suggesting a possible direct interaction with steroid hormones [3,4,9,10,11,12]. Thus, the aim of this review will be to summarize the putative effect of tadalafil on ARO activity and on steroid hormone receptor(s) expression by using both in in vitro and in vivo different cellular models and to translate them into possible clinical applications.

## 2. Tadalafil and Adipose Tissue: Clinical and Preclinical Observations

Serendipity, a clinical observation coming from our pilot study, suggested that the administration of tadalafil in men with ED was associated with an increase of the serum testosterone/estradiol (T/E2) ratio, mainly due to a significant reduction of E2 levels [4]. In that pilot study, where both lean and obese subjects were investigated, no clear-cut explanation for the observed serum E2 decrease was found or why this effect might have occurred. Even if a possible variation of the ARO activity was hypothesized, in that moment, any potential effects of PDE5i on ARO were unexplored. So, on the basis of our previous pilot study, we successively investigated the potential role of chronic tadalafil administration in men affected by ED and metabolic disorders, and we demonstrated its ability to improve insulin secretion and reduce visceral fat mass [10]. Similar observations have been reported by other published clinical studies [13] and by clinical trials actually registered on ClinicalTrials.gov which investigate the possible role of tadalafil on metabolism and body composition disorders (Table 1). Importantly, successive preclinical studies confirmed our clinical observation in men [9,14]. In a mouse model of diet-induced, insulin-resistant, chronic treatment with the PDE5i sildenafil caused a significant improvement in insulin sensitivity [14]. This result was promptly investigated in a successive animal study by controlling the expression and activity of PDE5 in different cell lines of rabbit adipose tissue (i.e., visceral vs. subcutaneous) in different metabolic conditions. This study demonstrated that tadalafil counteracted high fat, diet-associated visceral adipose tissue alterations by restoring insulin sensitivity and prompting preadipocytes differentiation towards a metabolically healthy phenotype where brown, fat-specific genes (such as uncoupling protein-1) are mostly expressed [9]. Apart from clinical and preclinical observations, controversial data confirming a role for PDE5 in adipocyte biology in vitro have been reported [3]. On the basis of our previous clinical observations, as confirmed by successive preclinical data, we subsequently found for the first time that PDE5 mRNA is present in human adipocytes and that selective PDE5 inhibition significantly stimulated ARO mRNA expression in mature adipocytes in vitro upon short-time exposure with a parallel increase in E2 concentrations in the supernatant [3]. E2 is synthesized by cytochrome P450-ARO, which converts androgens into estrogens; indeed, changes at the level of estrogen biosynthesis are closely related to modifications in the transcription of ARO and may exert an important role in the prevention of cardiovascular and metabolic disease [3]. These findings showed, for the first time, that acute PDE5 inhibition was able to increase ARO mRNA expression, which should reflect positive, anti-adipogenic effects. Finally, another study confirmed that even human skeletal muscle cells are a target tissue of tadalafil, and experimental data support its pharmacological actions on modulating glucose metabolism through a direct control on insulin signaling, on improving sex hormones profile and body composition, and on ARO expression, as well as on increasing exercise capacity due to its cardiovascular and vasodilatory effects, as demonstrated by in vitro and in vivo studies [12]. However, to date, neither the molecular mechanisms underlining the interactions between tadalafil, steroid hormones and skeletal muscle metabolism and differentiation are clear, nor are the mechanisms by which PDE5i might positively influence hormone metabolism and physical activity. These clinical and preclinical results led us to conclude that, at least in theory, a dose-dependent, tadalafil-related stimulation of ARO activity could positively modulate the serum T/E2 ratio in vivo during chronic treatment with tadalafil, and this might represent a possible mechanism influencing fat-mass content and its hormonal functions [3]. Thus, we can speculate that the stimulation of the NO/cGMP signal transduction system through a PDE5 blockade can provide a new, effective and reliable ‘target’ for deranged adipose tissue pathways, suggesting its potential role in the treatment of some forms of abdominal fat accumulation and mild obesity (Figure 1).

## 3. Tadalafil and Bone: Preclinical Observations

Over the last two decades, many physiological studies have demonstrated a tight association between NO, PDEs and bone cells homeostasis [15]. Preclinical studies showed that mice lacking NOS presented an osteoporotic phenotype, and both preclinical and clinical studies showed that the treatment with NO donor drugs improved bone mineral density and reduced fracture risk [16,17,18,19]. Moreover, murine studies showed that a high function of the cGMP–dependent protein kinase G (PKG), which is the downstream target of NO and is inactivated by PDEs, determines a high bone mass phenotype [20,21], suggesting a key role for the NO-cGMP-PKG axis in the regulation of bone remodeling, and suggesting that the inhibition of PDEs may represent a protective factor against bone loss. In a recent and interesting study, Kim at al. demonstrated that the expression of PDE5A was significantly higher in the bones of old mice than those of young ones, as was the expression of the molecular components of the NO-cGMP-PKG axis, and they demonstrated that the administration of tadalafil and vardenafil increased bone mass through central and peripheral actions. In particular, they observed that PDE5i acts directly on osteoblasts by modulating the expression of specific genes involved in the osteoblastogenesis (*Ogn* and *Bsp* were up-regulated; *Bmp2* was suppressed) in a time- or dose-dependent manner, or both [15]. Finally, since PDE5A-positive sympathetic neurons were found to innervate bone [22,23], Kim et al. studied the osteoblast precursors of tadalafil- and vardenafil-treated mice, and they demonstrated that both drugs suppressed specific sympathetic neuron-regulated genes involved in osteoblast precursor proliferation (*Per1*, *Per2*, *Bmal1*, *Myc* and *Ccnd*), while they exerted a direct anabolic action favoring the mineralization process and new bone formation [15]. Thus, since 47% of men over 50 years are clinically positive for osteopenia [24], the authors concluded that the use of PDE5i in aging men, to treat ED and/or BPH associated symptoms and low urinary tract symptoms (LUTS), may protect them from bone loss [15]. Moreover, in a rat model of glucocorticoid-induced osteoporosis, markers of oxidative stress and bone atrophy were significantly reduced by treatment with the PDE5is, zaprinast and avanafil [25], while PDE5 inhibition with vardenafil, udenafil, and tadalafil increased bone angiogenesis and bone formation rate and improved oxidative stress markers and resorption markers in osteoporotic ovariectomized rats [26]. Similar observations were found by Pal et al. in a model of mouse calvarial osteoblasts treated with sildenafil and vardenafil, which increased surface referent bone formation and serum bone formation marker P1NP and the expression of vascular endothelial growth factor and its receptor 2 in bones and osteoblasts, alongside increased skeletal vascularity [27]. However, neutral results came from an interesting study by Wang et al. conducted on the primary cultures of chondrocytes from newborn rat epiphyses [28]. The authors demonstrated that this cell line highly expressed PDE5 and that a short-term tadalafil treatment in growing rats at doses comparable to those used in children with pulmonary arterial hypertension has neither obvious beneficial effects on long bone growth nor any observable adverse effects on growth plate structure and trabecular and cortical bone structure [28]. Finally, we demonstrated that human osteoblasts (SAOS-2) express significant levels of both PDE5 mRNA and protein and that their exposure to increasing concentrations of tadalafil [10(-8)-10(-7) M] decreased PDE5 mRNA and protein expression. Further, in this cellular model, we demonstrated that tadalafil inhibited ARO mRNA and protein expression, leading to an increase in T levels in the supernatants, and that, interestingly, tadalafil increased total AR mRNA and protein expression and decreased ERα with an increased ratio of AR/ER, suggesting preferential androgenic vs. estrogenic pathway activation [29]. These consistent results confirm that tadalafil decreases ARO expression and increases AR protein expression in human SAOS-2 cells, strongly suggesting a new control of steroid hormones pathway by PDE5i, and representing the first evidence of translational actions of PDE5i on AR, which leads us to hypothesize a growing relevance of this compound in men with ED and prostate diseases under long-term treatment with tadalafil for sexual rehabilitation [29]. On the other hand, another study conducted on the cell lines of osteoblasts (UMR106 cells) demonstrated that the inhibition of PDE5 promotes ARO-mediated estrogen biosynthesis by the activation of the cGMP/PKG/SHP2 pathway, suggesting a potential role of cGMP and the PDE5 blockade in the regulation of estrogen biosynthesis in bone tissue [30].

Unfortunately, no targeted results from clinical studies are available at present. Certainly, to confirm the hypothesis coming from preclinical observations, ad hoc-designed clinical studies having as a primary endpoint the effects of PDE5i on bone metabolism and skeletal preservation (Figure 1), in aging males, are deemed necessary.

## 4. Tadalafil and Prostate Cancer: Clinical and Preclinical Observations

PDE5is are largely used as daily treatment for BPH-related, LUTS-lowering, spontaneous contractility of the glands, thereby reducing the muscle tone of the genitourinary tract [31]. The reported up-regulation of PDE5 in hyperplastic human prostates could provide a rationale for treating patients with LUTS/BPH with/without ED [32]. Furthermore, different clinical trials have been performed, reporting the use of PDE5i benign prostatic hyperplasia, but, to date, very scarce data on PCa are available (Table 2). In contrast, a specific inhibitory growth pattern on prostate tissue has not been clearly documented. Immunohistochemical studies have shown PDE5 immunolocalization mainly in the fibromuscular stroma and vascular (endothelial and stem) cells in the rat and human prostate of BPH subjects [33], as well as in the glandular and subglandular areas of human prostate cancer (Pca) patients [34].

In adult males, PCa is a leading cause of death. Often, castration-resistant prostate cancer (CRPC) has a lower therapeutic response to conventional chemotherapy [35], and the expression of PDE5 and cGMP-signaling pathway in normal and cancerous prostate tissues and their possible involvement in carcinogenesis still remains controversial. Although PDE5 immunolocalization studies in prostate adenocarcinomas have not been exhaustively reported in the scientific literature, PDE5is are largely used after oncological curative treatments for PCa. Bisegna et al. [36] recently studied the expression of PDE5 in human healthy and pathological prostate sections, and, interestingly, they found PDE5 overexpression in the epithelial compartment, but not in the stromal cells, of 22% of PCa samples compared to normal (8%) or hyperplastic samples (11%). Such positivity was not correlated with the Gleason grading system. The authors concluded that the lack of correlation with the Gleason score suggests that this enzyme is not correlated with tumor aggressiveness; however, a role of PDE5 in the minimal residual disease of PCa cannot be excluded [36]. These results prompted us to investigate the effects of tadalafil on the expression of AR and ARO, and its potential impact in modulating the antiproliferative activity of androgen deprivation therapy (ADT) in human PCa, androgen-sensitive PCa cell line (LnCAP) cells [37]. We demonstrated for the first time that tadalafil can modulate AR expression in prostate cancer cells in vitro, and it can induce the stabilization and reduce the degradation of AR [37]. Interestingly, this effect could lead to a potential anti-cancer action of tadalafil by enhancing the therapeutic effect of ADT. The acute exposure of LNCaP to tadalafil did not affect cell viability and proliferation rate. Tadalafil, without affecting either the metabolism or proliferation of PCa cells, up-regulates AR protein expression and transcriptional activity [37]. AR is the classical target for PCa prevention and treatment, but more recently, estrogens and their receptors have also been involved in both tumor development and progression. Local estrogen signaling mechanisms are required for prostate carcinogenesis and tumor progression [38], but the role of estrogens in the pathophysiology of PCa is not well established. The hypothesis of the role of estrogens in the regulation of prostate growth in men was confirmed by Bonkhoff [39]. In fact, the estrogen receptor beta (ERβ) is most prevalent in the human prostate, while the expression of the estrogen receptor alpha (ERα) is restricted to the basal cells of the prostatic epithelium and the stromal cells. In high-grade prostatic intraepithelial neoplasia, the ERα might be up-regulated, while a partial loss of the ERβ might occur, suggesting its potential action as a tumor suppressor. The ERβ is generally retained in hormone naïve and metastatic PCa, but it is partially lost in castration-resistant disease [40]. Studies conducted in hypogonadal ARKO mouse models, when exposed to E2, have demonstrated the pivotal role of estrogen in the proliferative response of the prostatic stroma and epithelium [41]. Finally, induction of prostate carcinoma requires the combined actions of both T and E2, since both androgens and estrogens could initiate changes in the prostate independently, but they cannot individually produce malignancy [42]. Moreover, in PCa cell lines, the AR antagonist bicalutamide (BCT) increased ARO expression and ERβ transcriptional activity; indeed, in CRPC, ARO expression was significantly increased in tumor samples. Our study carried on with LnCAP cells and demonstrated for the first time that chronic exposure (48 h) to BCT produced a significant increase in ARO mRNA, which was reverted by co-treatment with tadalafil [37]. The inhibition of PDE5 has been shown to induce anticancer effects [43] both in pre-clinical [44,45] and clinical experiences [46]. Indeed, anastrozole and selective aromatase inhibitors had been proposed for the treatment of men with advanced prostate cancer, but currently, results are still inconsistent [47]. Attia and Ederveen demonstrated that a high expression of ERβ in PCa cells increases cell apoptosis and decreases cell proliferation, exerting a potential pharmacological role in neoplastic lesions [48]. We herein speculate that the local increase in estrogen levels might activate ERβ intracellular pathway and that chronic exposure to BCT may induce a loss of ERβ due to the induction of ARO. Tadalafil potentiates the antiproliferative activity of BCT in LnCAP cells [37], and co-treatment with tadalafil was able to block these effects, thus leading us to hypothesize the maintenance of androgen responsiveness to anti-androgen therapy (Figure 1). Finally, in a study performed by Hankey et al., in prostate cancer cell lines, the authors demonstrated that the treatment with PDE5i at clinically relevant concentrations did not induce variation in the proliferation, colony formation or migration phenotypes, even when cells were co-treated with a stimulator of cGMP synthesis to facilitate cGMP accumulation upon PDE5 inhibition. Surprisingly, supraclinical concentrations of PDE5i counteracted proliferation, colony formation and the migration of prostate cancer cell models. These findings provide tumor cell-autonomous evidence in support of the field’s predominant view that PDE5is are safe adjuvant agents to promote the functional recovery of normal tissue after prostatectomy, but the authors did not rule out the potential cancer-promoting effects of PDE5i in the more complex environment of the prostate [49]. Even if it is known that higher PDE5A expression may predict survival in human PCa in the clinical context (data from www.proteinatlas.org, accessed on accessed 30 March 2022), whether its functional inhibition may determine better survival in CRPC in humans is not known.

## 5. Conclusions

It is known that the PDE5 enzyme is widely distributed in human tissues, including fat, bone and the genitourinary tract, and that it is involved in the pathophysiological process of numerous diseases, including cancers. In recent decades, numerous pharmaceutical compounds that selectively inhibit the catalytic activities of PDEs have been developed for the treatment of various andrological disorders, and it has also been demonstrated that PDE5i can exert protective effects in different clinical conditions, apart from ED, such as myocardial infarction, endothelial dysfunction, platelet aggregation, insulin resistance and, finally on skeletal muscle functions. Since these considerations, in this review, we have summarized the putative effect of tadalafil on ARO activity and on steroid hormone receptor(s) expression by analyzing both in vitro and in vivo different cellular models to translate them into possible clinical applications. We conclude that tadalafil can operate as selective modulator of ARO expression and functions, depending on the tissue and organ involved, and it has stimulatory effects on adipocytes and PCa cell lines and inhibitory actions on bone tissues. In other words, we believe that the clinical relevance of this interaction may yield some clinical benefits, i.e., it may reduce abdominal fat mass plus improve insulin sensitivity, and increase the testosterone/estradiol ratio, which may exert a protective effect on trabecular bone independent of gender. This dimorphism needs further investigation based upon sex differences in the clinical context. Finally, its monomorphic action on facilitating the translocation of AR into the nucleus independently from the cellular system studied means that the daily use of this drug may determine changes in cell machinery, switching them onto responsive to androgen/antiandrogen treatments depending on the clinical context considered, i.e., late-onset hypogonadism [50] or CRPC, and thus, potentially impacting clinical outcomes. 

## Figures and Tables

**Figure 1 ijms-23-04191-f001:**
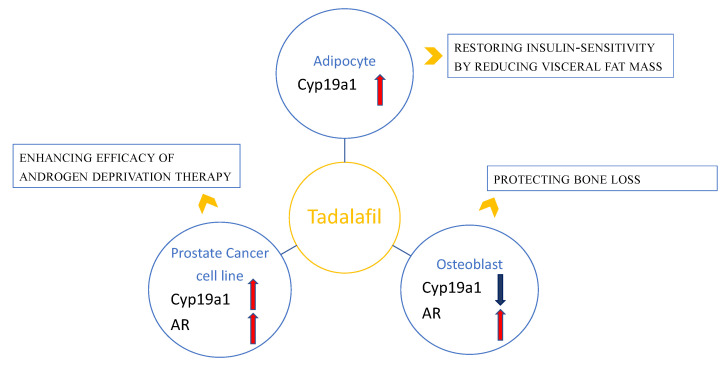
Schematic representation of the relationship between tadalafil and steroid hormone pathways in adipocyte, osteoblast and prostate cancer cell lines and the clinical translational perspective of tadalafil actions. Tadalafil may modulate the expression of molecules such as Cyp19a1 and the androgen receptor (AR). In particular, tadalafil increases the expression of AR in osteoblast and prostate cancer cell lines and decreases Cyp19a1 expression in osteoblasts, whereas it increases the expression of Cyp19a1 in adipocyte and prostate cancer cell lines.

**Table 1 ijms-23-04191-t001:** List of studies on PDE5 inhibitor tadalafil, body mass and insulin resistance/sensitivity at www.clinicaltrials.gov (accessed 30 March 2022).

Indication	Outcomes	Status	ClinicalTrials.gov Identifier
Obesity	Improves: Fasting and post-prandial glucose Insulin secretion Insulin sensitivity AUC Glucose AUC Insulin BW BMI WC BF Cholesterol (total and LDL fraction) BP	Phase 3 (completed)	NCT02595684
Cardiovascular Disease Insulin Resistance Glucose Intolerance Obesity	Improves: endothelial function insulin sensitivity Matsuda Disposition Index	Phase 3 (completed)	NCT01444651
Metabolic Syndrome	Improves: β-cell function insulin sensitivity	Phase 3 (completed)	NCT00750308
Obesity	Improves: BMI Variation in lean and fat mass	Phase 3 (completed)	NCT02554045

Legend: AUC: area under the curve; BW: body weight; BMI: body mass index; WC: waist circumference; BF: body fat; LDL: low density lipoprotein; BP: blood pressure, ED-METABOLIC: erectile dysfunction (ED) in men with ED and metabolic syndrome.

**Table 2 ijms-23-04191-t002:** List of studies on PDE5 inhibitor tadalafil, benign prostatic hyperplasia and cancer at www.clinicaltrials.gov (accessed on 30 March 2022).

Indication	Outcomes	Status	ClinicalTrials.gov Identifier
BPH	Improves: IPSS score CGI-I scale PGI-I scale	Phase 3 (completed)	NCT01460342
	Improves: PdetQmax Urinary Flow Rate Bladder function	Phase 3 (completed)	NCT00386009
	Improves: AUC Cmax Tmax	Phase 3 (completed)	NCT01183650
	Improves: IPSS score BPH index CGI-I scale PGI-I scale	Phase 3 (completed)	NCT00827242
	Improves: IPSS score Treatment-emergent Dizziness PVR Uroflowmetry	Phase 3 (completed)	NCT00848081
	Improves: IPSS score Quality of Life Bladder function	Phase 3 (completed)	NCT00783094
	Improves: IPSS score Quality of Life BII index Erectile function Bladder function	Phase 3 (completed)	NCT00970632
	Improves: IPSS score Quality of Life BII index Erectile function Bladder function	Phase 3 (completed)	NCT00861757
	Improves: IPSS score BII index Bladder function	Phase 3 (completed)	NCT00540124
	Improves: IPSS score BII index Bladder function Erectile function Quality of Life	Phase 3 (completed)	NCT00384930
	Improves: RI index CPI index	Phase 3 (completed)	NCT01152190
	Improves: CGI-I PGI-I IPSS score	Phase 3 (completed)	NCT02431754
	Improves: IIEF-EF Sexual function IPSS score	Phase 3 (completed)	NCT00855582
	Improves: IIEF-EF Sexual function IPSS score	Phase 3 (completed)	NCT01937871
	Improves: IIEF-EF CGI-I IPSS score	Phase 3 (completed)	NCT01139762
	Improves: PVR PV PSA IPSS score	Phase 1 (recruiting)	NCT04947631
	Improves: Uroflowmetry parameters PSA IPSS score	Phase 3 (completed)	NCT00547625
	Improves: IPSS score	Phase 3 (completed)	NCT03246880
	Improves: AUClast Cmax	Phase 3 (completed)	NCT02352311
	Improves: LUTS/BPH symptoms Pressure flow parameters Inflammatory markers Prostatic microcalcifications Metabolic profile Erectile function	Phase 3 (completed)	NCT02252367
	Improves: Pressure flow parameters Urinary symptoms Therapy compliance	Phase 3 (completed)	NCT04383093
Prostate cancer	Improves: Erectile function Sexual function	Phase 3 (completed)	NCT00931528
	Improves: Erectile function Clinical/biochemical recurrence of cancer	Unknown	NCT00906269
	Improves: Erectile function Sexual function	Phase 3 (completed)	NCT02103088

Legend: BPH: benign prostatic hyperplasia; IIEF: International Index of Erectile Function; ED: erectile dysfunction; IPSS: International Prostate Symptom Score; CGI-I: Clinician Global Impression of Improvement; PGI-I: Patient Global Impression of Improvement; PdetQmax: detrusor pressure at peak urinary flow rate; AUC: area under the concentration curve for tadalafil and metabolite; Cmax: concentration maximum of tadalafil; Tmax: time to concentration maximum of tadalafil and metabolite; BPH: benign prostatic hyperplasia impact index; PVR: postvoid residual volume; BII: Benign Prostatic Hyperplasia Impact Index; RI: Arterial Resistive Index; CPI: color pixel intensity; IIEF-EF: erectile function (IIEF-EF) domain score; PV: prostate volume; PSA: prostate specific antigen; LUTS: lower urinary tract symptoms.

## Data Availability

Not applicable.

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
