# Peer review of "Tadalafil and Steroid Hormones Interactions in Adipose, Bone and Prostate Tissues: Focus on Translational Perspectives"

_ijms, 2022, doi:10.3390/ijms23084191_

Round 1

Reviewer 1 Report

The authors write a review article on the physiological and pathological role of PDE5 in adipocytes, bone and prostate, especially its association with sex hormones and their receptors. It also describes the basic and clinical effects of PDE5 inhibitors.

The authors mix their own research findings, and this paper is a concise and excellent review.

Author Response

We thank the reviewer for her/his comments

Reviewer 2 Report

This work by Greco and co-workers summarizes basic and clinical results concerning the potential use of PDE5i for the prevention/treatment of obesity, bone loss and prostate cancer.

The topic is interesting and worth reviewing, since there are not so many recent reviews on this topic, however some points need to be better addressed to reach the standard for publication in IJMS.

I really hope that my comments and suggestions can be appreciated by the authors and that they can be useful to improve the overall quality of the work.

Overall, I found the reading a bit heavy and full of vague and repeated concepts. I suggest lightening the text by removing redundant concepts and giving more logic to the sections. It is not clear why the authors frequently move from basic to clinical results without following a logical thread. I suggest at least to start from preclinical evidences in each section or to dedicate a subsection to preclinical studies. Section 5 and relative subsections seem to be redundant in the present form: most information have already been mentioned in the previous sections.

The abstract should be revised substantiating the conclusions: statements such as “different clinical conditions” should be avoided.

In the introduction the authors state that the phosphodiesterase superfamily is composed of “11 isoforms”. This information is inaccurate since known PDEs isoforms are much more than 11. Please use the opportune terminology “11 families”.

A strong revision and implementation of references is mandatory. Often a concept does not correspond to the appropriate reference (e.g lane 32 ref. 1 refers to adipogenesis but statement refers to cancer; lane 36 ref.4 refers to diabetes but authors are referring to tumors; lane 40 ref.5 refers to rabbits but authors are focusing on ED patients) and too many sentences are not properly supported by any reference (e.g. lanes 29, 42, 136, 186, 193, 204 and section 5). Moreover some important works on the topic are missing and should be included and possibly commented:

- Wisanwattana W. et al., Front. Endocrinol., 2021 doi: 10.3389/fendo.2021.636784

- Hankey W. et al., Translational Oncology, 2020 doi: 10.1016/j.tranon.2020.100797

- Campolo et al., Curr. Opin. in Pharmacol., 2021 doi: 10.1016/j.coph.2021.08.007

- Wang L. et al., Am. J. Physiol. Endocr. Met., 2018 doi: 10.1152/ajpendo.00130.2018

Figure 1 should be revised: there is no correspondence between figure and its description in figure legend in osteoblast. It is also unclear why authors use singular for osteoblast and plural for adipocytes and prostate is referring to organ instead cells. A standardization will be very appreciated.

The statement “chronic exposure (48hrs)” is bizarre and unconventional.

At least a mention to clinical trials on PDE5i and metabolic, bone and prostate disorders should be included.

The manuscript needs to be carefully edited for English language: the authors should seek editorial assistance to improve it.

Author Response

(The authors gave the same response as above.)

Reviewer 3 Report

Dear aurhors,

In this review, the authors summarize the translational potential of tadalafil in adipose tissue (or in insulin insensitivity), bone formation (osteoporosis), and prostate magnancy. In my opinion, I will suggest the authors in several part:

  1. For monitoring mody mass and insulin sensitivity, tadalafil has involved in several clinical trials and some of which has result (NCT01444651, NCT02595684). If the authors can form a table about the clinical trials using PDE5 inhibitors (or particularly tadalafil) in insulin sensitivity, the therapeutic potential of tadalafil can be further reinforced.
  2. Similar situation, PDE5 inhibitor (not tadalafil) had several clinical trials in osteoporosis. I suggest the authors summarizing them in this review.
  3. As I survey the prognostic correlation of PDE5 to prostate adenocarcinoma, I find that PDE5 seems a favorable factor of prostate adenocarcinoma (https://www.proteinatlas.org/ENSG00000138735-PDE5A/pathology/prostate+cancer#imid_3386556, https://www.cbioportal.org/results/comparison?cancer_study_list=prad_mich%2Cprad_su2c_2019%2Cprad_su2c_2015%2Cprad_mcspc_mskcc_2020%2Cnepc_wcm_2016%2Cprad_broad_2013%2Cprad_broad%2Cprad_cpcg_2017%2Cprad_fhcrc%2Cprad_cdk12_mskcc_2020%2Cprad_mskcc%2Cprad_mskcc_2014%2Cprad_p1000%2Cprad_eururol_2017%2Cprad_tcga_pub%2Cprad_tcga%2Cprad_tcga_pan_can_atlas_2018%2Cprad_mskcc_cheny1_organoids_2014%2Cprostate_dkfz_2018%2Cprad_msk_2019%2Cprad_mskcc_2017%2Cprad_msk_stopsack_2021%2Cmpcproject_broad_2021&Z_SCORE_THRESHOLD=2.0&RPPA_SCORE_THRESHOLD=2.0&profileFilter=mutations%2Cfusion%2Ccna%2Cgistic&case_set_id=all&gene_list=PDE5A&geneset_list=%20&tab_index=tab_visualize&Action=Submit&comparison_subtab=survival). The authors need to confirm this situation.
  4. Due to absent reports of tadalafil treated prostate adenocarcinoma, the authors use the results from other two PDE5 inhibitors to demonstrate the therapeutic potential of tadalafil in prostate adenocarcinoma. However, Chang et al. demonstrated that sildenafil and verdenafil had topoisomerase II inhibitor activity but tadalafil hadn't (10.3389/fonc.2018.00681). In this case, I think using sildenafil  and verdenafil to inference the therapeutic potential of tadalafil is inappropriate. 
  5. The authors described that PDE5 destributed in fibromuscular and stromal tissue in prostate gland. However, most prostate cancer patients are prostate adenocarcinoma, not fibrosarcoma. I suggest the authors emphasize the linkage of PDE5 to prostate adenocarcinoma. 

Author Response

(The authors gave the same response as above.)

Round 2

Reviewer 2 Report

Authors reply satisfied my requests.

Their work now meets the criteria for publication in IJMS

Author Response

(The authors gave the same response as above.)
